# The Fate of Transplanted Olfactory Progenitors Is Conditioned by the Cell Phenotypes of the Receiver Brain Tissue in Cocultures

**DOI:** 10.3390/ijms21197249

**Published:** 2020-09-30

**Authors:** Pourié Grégory, Akchiche Nassila, Millot Jean-Louis, Guéant Jean-Louis, Daval Jean-Luc, Bossenmeyer-Pourié Carine

**Affiliations:** 1Inserm UMR1256/NGERE, Faculté de Medecine, 9 Avenue de la Forêt de Haye, Université de Lorraine, F-54000 Nancy, France; nassila.akchiche@univ-lorraine.fr (A.N.); jean-louis.gueant@univ-lorraine.fr (G.J.-L.); jean-luc.daval@univ-lorraine.fr (D.J.-L.); carine.pourie@univ-lorraine.fr (B.-P.C.); 2Laboratoire de Neurosciences, Place Leclerc, Université de Franche-Comté, F-25000 Besancon, France; jean-louis.millot@univ-fcomte.fr

**Keywords:** cellular therapy, olfactory progenitors, circuit integration, cortex, hippocampus

## Abstract

Among the numerous candidates for cell therapy of the central nervous system (CNS), olfactory progenitors (OPs) represent an interesting alternative because they are free of ethical concerns, are easy to collect, and allow autologous transplantation. In the present study, we focused on the optimization of neuron production and maturation. It is known that plated OPs respond to various trophic factors, and we also showed that the use of Nerve Growth Factor (NGF) allowed switching from a 60/40 neuron/glia ratio to an 80/20 one. Nevertheless, in order to focus on the integration of OPs in mature neural circuits, we cocultured OPs in primary cultures obtained from the cortex and hippocampus of newborn mice. When dissociated OPs were plated, they differentiated into both glial and neuronal phenotypes, but we obtained a 1.5-fold higher viability in cortex/OP cocultures than in hippocampus/OP ones. The fate of OPs in cocultures was characterized with different markers such as BrdU, Map-2, and Synapsin, indicating a healthy integration. These results suggest that the integration of transplanted OPs might by affected by trophic factors and the environmental conditions/cell phenotypes of the host tissue. Thus, a model of coculture could provide useful information on key cell events for the use of progenitors in cell therapy.

## 1. Introduction

During the last few decades, cell therapy for tissue repair has brought great hope in numerous diseases because stem cells from various origins can differentiate into defined phenotypes [1]. Concerning the central nervous system (CNS), different approaches and techniques have been developed for cell replacement, such as systemic injections of stem cells into the blood or direct brain grafts [2,3]. Some of these cell therapy models for brain repair gave good results in functional restoration [4,5,6,7]. Spinal-cord injuries can also be successfully repaired with various stem/progenitor cells. Nevertheless, authors pointed out the necessity for a better understanding of the mechanisms involved in these successful engraftments [8,9,10]. However, in the majority of cases, it is difficult to avoid donor death since stem/progenitor cells need to be collected from specific tissues such as the forebrain [11,12] or midbrain [13,14]. In order to avoid invasive collection, other sources were used such as stem cells derived from aborted embryos [6], umbilical cord blood [15,16], or bone marrow [17,18,19]. However, important ethical problems need to be addressed before considering any of these approaches for clinical applications [20,21,22].

Accessible sources of progenitor cells are very scarce in adults. Some authors reported that the skin could be a good candidate [23], but the olfactory epithelium seems to be an interesting source of progenitors because it regenerates frequently [24,25] and contains stem-like/progenitor cells [26,27,28]. Thus, olfactory progenitors (OPs) can be easily collected using a noninvasive method [29], extended as spheres in vitro, and they are suitable for efficient transplantation [30,31,32]. A study reported that OPs could be used not only for strategies of cell replacement, but also for their trophic factor secretion (i.e., Brain-Derived Neurotrophic Factor, BDNF) to rescue axotomized neurons [33]. Nevertheless, progenitor cell therapy in neurodegenerative conditions and trauma remains the main objective, since such undifferentiated cells are able to respond to various differentiation factors [34,35,36]. Despite interesting studies on OP transplantation [37,38], no data are available concerning OP viability and integration into the mature neuronal network, which constitutes a critical point to be elucidated for cell therapy.

In the present study, we investigated the multiplication of adult olfactory progenitors and their ability to form neurospheres, as previously demonstrated [39], but we focused on the different conditions in which these cells differentiate in vitro, in order to describe whether they can survive and integrate a formed network with both neurons and astrocytes [40]. To date, even if some functional improvements or recovery were observed after grafting, very little is known concerning the conditions in which exogenous progenitors can integrate a mature circuit, damaged or not. Therefore, we developed a coculture model in order to address some questions concerning the crosstalk between grafted and host cells, and to study the fate of transplanted progenitors in mature tissues.

## 2. Results

### 2.1. Cultured Olfactory Progenitors

Cultured cells were observed under a 20× magnification every day. Isolated cells were observed the first day of culture (Figure 1A), but proliferation led rapidly (5 days) to small aggregates of 2–6 cells (Figure 1B). After 10 days of culture, cells formed larger aggregates (Figure 1C), and, after 2 weeks, they grew as neurospheres (Figure 1D).

Cell viability was monitored for 25 days in cultures of OPs. Both used methods (i.e., dimethylthiazol-diphenyl tetrazolium salt, MTT and trypan blue exclusion) showed the same profile during the culture (Figure 2). A significant threefold decrease in the cell viability and number of cells (9000 to 3000 per 100 µL, F (7, 24) = 48.1; *p* < 0.0001) was recorded during the first week of culture. Then, a stabilization was observed during the two following weeks, with a slightly varying number of cells from 3000 to 4000 per 100 µL. Finally, the cell viability reached the same level as that measured at the beginning of the culture (Figure 2A). The number of living cells was increased by 33% compared with the beginning of the study, showing a significant increase from day 14 to day 25, from 4000 to 13,000 cells (F (7, 24) = 48.1; *p* < 0.0001, Figure 2B).

### 2.2. Cell Phenotypes in Olfactory Epithelium Cultures

In order to monitor the phenotype of cells in the cultures of olfactory epithelia, we followed their maturation during the first three weeks by immonocytochemistry. Antibodies were chosen to focus on the balance between immature and mature cells: nestin for immature cells, NeuN for mature neurons, and GFAP for mature astrocytes.

Figure 3 shows that only mature cells (i.e., neurons and glial cells) were found at the beginning of culture, where immature cell phenotypes were undetected. At the end of the first week, glial cells still represented half of the total number of counted cells, and the mature neuron phenotype started decreasing while the nestin immature marker started increasing to 10% of total cells. Finally, from the second to the third week of culture, all mature cells decreased to reach a low level of 10% to 20%, and, at the same time, an increase in the number of immature cells was observed to a level as high as 60% by day 21.

### 2.3. Differentiation of Olfactory Progenitors

In order to investigate whether cells dissociated from neurospheres were able to differentiate properly in the neuronal lineage, we cultured these cells in small dishes with or without Nerve Growth Factor (NGF). The use of 100 neurospheres allowed plating 7500 ± 980 living cells. The cell morphology was followed using a phase microscope for 10 days.

Cell phenotype was studied after 10 days of adherent condition in both culture conditions. Qualitatively, both media allowed cells to differentiate into neurons or astrocytes, shown as a function of immunoreactivity to NeuN and GFAP, respectively (Figure 4). Moreover, progenitors engaged in neuronal differentiation also showed a synaptic process, since they coexpressed markers of neuronal cytoskeleton (Map-2) and synaptic proteins (Synapsin-I) (Figure 4C).

Cell counting showed that, even if both culture conditions led to significantly more mature neurons than glial cells, the presence of NGF in the medium promoted the percentage of neurons (83% with NGF versus 64% without NGF) (Figure 4D).

### 2.4. Viability of Primary Cultures and Cocultures

After six days, primary cultures of the cortex and hippocampus were tested for the proportions of mature neurons (NeuN-positive) and mature astrocytes (GFAP-positive) (Figure 5). The results showed that both cortex and hippocampus cultures contained significantly more neurons than glial cells. However, the proportion of neurons in the hippocampus primary cultures was significantly higher than in the cortex (85% vs. 70%).Concerning cell viability, the MTT method used on whole-dish extracts (Figure 6) showed that primary cultures of the cortex and hippocampus were stable from day 6 to day 12 after plating, since no significant differences were found according to the time of culture. Nevertheless, cortex cultures showed significantly higher viability on day 6 and day 12 (F(1-14) = 9.5 to 9.7, *p* < 0.008).

When dissociated immature olfactory progenitors were plated on top of primary cultures on day 6, higher cell viability was observed at all measurement points (from day 8 to day 12) regardless of the host circuit (i.e., cortex or hippocampus), due to the addition of new cells. However, only the addition of OPs on cortex cultures induced a significant increase in the global cell viability of the coculture when compared to the cortex control on days 8 and 10 (F(3-28) = 2.8 to 3.9, *p* < 0.05). The addition of OPs on the hippocampus culture did not induce such an increase, indicating a lower mortality of OPs on the glial-rich cortex host than on the hippocampus host.

### 2.5. Olfactory Progenitor Differentiation in Cocultures

Phenotypes of olfactory progenitors previously treated with BrdU were studied on day 12 of culture, which corresponds to day 6 after immature progenitor plating. Data showed that only the mature neuronal marker could be found in differentiated progenitors (coexpression of BrdU and Map2, Figure 7). After investigation of several fields of cells randomly chosen in three different wells on day 12, the glial marker was not present in cells with BrdU-labeled nuclei, indicating that olfactory progenitors should respond to factors existing in the host culture and leading preferentially to the neuronal lineage in these conditions. Moreover, some BrdU-labeled cells coexpressed the cytoplasmic synapsin I, indicating that a mechanism of functionality was at least engaged (Figure 7B).

## 3. Discussion

Our results show that olfactory progenitors (OP) expanded as neurospheres are able, like embryonic cells [6,41,42], to differentiate after dissociation in at least two brain lineages, i.e., neurons and astrocytes [43,44]. This confirms that progenitors from the olfactory epithelium are multipotent [45], an essential quality for potential use in cell therapy.

When OPs were plated alone, they were able to differentiate into the neuronal lineage after NGF induction, as also shown by others in vivo [46]. Thus, the cellular elongation (i.e., axon formation), revealed by Map-2 immunoreactivity, and the connection process, revealed by synapsin-I immunoreactivity, suggest that they have the potential to form an organized circuit by themselves [47].

In addition, coculture results showed that OPs can differentiate rather than proliferate while they integrate with a mature circuit with both neuronal and glial cells, as previously reported for other stem cells [48]. This is in accordance with recent studies indicating that OPs are able to respond to external factors such as the Vascular Endothelial Growth Factor (VEGF) or the Brain-Derived Neurotrophic Factor (BDNF) for proliferation, viability, and migration [49,50].

In our culture conditions, it appeared that OP displayed better integration and viability within cortex primary cultures. Compared to the hippocampus, cortex primary cultures were enriched in glial cells. This suggests that glial cells may enhance the integration, survival, and differentiation of progenitors. This is in accordance with Brakho et al. (2006) [51], who showed that progenitors plated on astrocytes presented a good fate, also suggesting that progenitors cocultured with assistant cells (i.e., protective, nutritive) can survive and differentiate in better conditions. In this respect, released trophic factors from astrocytes might modulate the outcome of immature plated cells and constitute a critical point for the success of any transplantation both in vitro and in vivo [11,51]. Indeed, during the last few decades, several repair trials were done, and authors pointed out the importance of providing a “graft-friendly” environment for stem cells [52,53,54], which was confirmed in our study.

Nevertheless, studies on OPs are still needed to improve expansion and specific differentiation. Actually, transplantation needs many immature cells, and, as already shown with other sources of stem-like cells, numerous successive passages in appropriate culture medium could provide enough cell density [15,55]. Feron et al. (2005) [56] showed the great potential of OPs for autologous transplantation; however, despite long-term monitoring after surgery showing no complications, very little is known concerning the survival and connection processes that allow good integration for OPs in mature circuitry. Regarding that specific point, an in vitro model of coculture could be very helpful to elucidate the molecular crosstalk when graft cells integrate with a mature tissue, as also shown with cells from other origins [57,58,59].

## 4. Materials and Methods

### 4.1. Animals

Experiments were performed on OF1 mice (Charles River, France) in accordance with the National Institute of Health Guide for the Care and Use of Laboratory Animals in an accredited establishment (Institut National de la Santé et de la Recherche Médicale, Unité 1256) according to the UE guidelines 2010-63-UE and to French governmental decree 2013-118. Mice were maintained under standard laboratory conditions on a 12 h light/dark cycle with access to food and water ad libitum, and environmental enrichment with wooden pieces. Since none of the animals used were kept alive after cell collection, no animal suffering was recorded and no analgesia was necessary.

For both adult (OP cultures) and newborn (primary cultures) mice, neuronal tissues were obtained after gas anesthesia using isoflurane diffusion. Then animals were killed by decapitation. A total of 10 adults and 20 newborns from three different litters were used.

### 4.2. Olfactory Progenitor Culture

Olfactory progenitors were obtained from the neuroepithelia of 10 adult (4–8 weeks old) mice. The olfactory epithelia were dissected and immediately placed in Hanks’ modified medium (Ca/Mg-free Hank’s Balanced Salt Solution (HBSS), 100 UI/mL penicillin, 100 µg/mL streptomycin, 15 mM hydroxyéthyl-pipérazine ethane sulfonic acid buffer (HEPES), 25 mM d-glucose). Tissue was then treated with dispase enzyme 2.4 UI/mL in HBSS medium in order to remove the chorion from the olfactory epithelium. The olfactory epithelium pieces were selected with a fine needle and placed in HBSS medium containing trypsin (0.5 mg/mL), and incubated at 37 °C for 5 mins. The reaction was stopped by the addition of trypsin inhibitor (1 mg/mL) and followed by mechanical cell dissociation. The cell suspension was centrifuged at 100× *g* for 8 min and cells were resolved in a proliferation medium (Neurobasal-A glutamine free, 200 µM l-glutamine, 1% B27 supplement, 50 µg/mL penicillin/streptomycin). The suspension was cultured at a density of 15 × 10^4^ cells/mL in a plastic dish at 37 °C, and Epidermal Growth Factor (EGF, 20 ng/mL) was added every 2 days. Under such conditions, isolated olfactory progenitors started proliferating to form neurospheres defined as round cell aggregates. For differentiation studies, the medium was supplemented on days 5, 10, and 15 with 50 nM BrdU (Sigma) in order to label olfactory progenitor cells.

### 4.3. Olfactory Progenitor Differentiation

After at least 4 weeks of culture, the differentiation of olfactory progenitors was evaluated in adherent conditions. A Pasteur pipette mounted on a micromanipulator was used to collect neurospheres one by one at 20× magnification. Cells were dissociated by accutase (Sigma) treatment (8 min at 37 °C), and finally centrifuged at 100× *g* for 8 min. The pellet was redispersed in the same medium (supplemented with 5% inactivated Fetal Calf Serum (FCS) on the first day of culture) and transferred into small six-square-well plates (Falcon, Becton Dickinson, Franklin Lakes, NJ, USA) precoated with poly-l-lysine to obtain a final density of 1000 cells/cm^2^. The medium was supplemented or not with 20 ng/mL NGF to evaluate the abilities of olfactory progenitors to differentiate in both conditions.

### 4.4. Primary Cultures

Primary cultured cells were obtained from the hippocampus or frontal cortex of 20 newborn mice (1 day old). Brains were removed and placed in culture medium previously equilibrated at 37 °C, consisting of a mixture of glutamine free Neurobasal-A, 200 µM l-glutamine, 1% B27 supplement, 50 µg/mL penicillin/streptomycin, and 5% inactivated FCS (Abcys, Paris, France). The hippocampus and frontal cortex were carefully collected at 10× magnification. Brain tissues were dissected free of meninges and gently dispersed in culture medium. After centrifugation at 100× *g* for 8 min, the pellet was redispersed in the same medium and passed through a 40 µm pore size nylon mesh.

Aliquots of the cell suspension were transferred to 12-well plates (Falcon, Becton Dickinson, Franklin Lakes, NJ, USA) precoated with poly-l-lysine to obtain a final density of 4 × 10^5^ cells/well. Cultures were then placed at 37 °C in a humidified atmosphere of 95% air/5% CO_2_. The following day, the culture medium was replaced with a fresh serum-free medium consisting of glutamine-free Neurobasal-A, 200 µM l-glutamine, 1% B27 supplement, 50 µg/mL penicillin/streptomycin, 20 mM KCl, fibroblast growth factor (2 ng/mL), and epidermal growth factor (10 ng/mL) (Sigma Chemical Co., St. Louis, MO, USA). After two additional days, the culture medium was renewed with serum-free medium in the absence of growth factors

### 4.5. Cocultures

To assess whether olfactory progenitors are able to integrate into a neural substrate, dissociated progenitors from neurospheres were transferred into 6 day old primary cultures of the cortex or hippocampus at a density of 1000 cells/cm^2^. The cocultures were maintained for 8 additional days, and cell viability and differentiation were monitored. The culture medium was renewed every 2 days with fresh serum-free medium.

### 4.6. Cell Viability

For the olfactory progenitor culture, two methods were used to assess cell viability. At various time points of culture (from 0 to 25 days), 200 µL was collected from culture dishes and immediately centrifuged at 100× *g* for 8 min, treated with accutase to avoid aggregates, and finally divided in two. The cell viability in one aliquot was determined by trypan blue exclusion (0.4% final concentration; further microscope counting using a calibrate grid). The cell viability in the second aliquot was monitored using a spectrophotometric method with tetrazolium salt MTT (Sigma) (see below).

For primary cultures, after 6 days, the culture medium was replaced with Dulbecco’s Modified Eagle Medium (DMEM), and freshly dissolved MTT (stock: 5 mg/mL in 0.1 M Phosphate Buffer Saline, PBS) was aseptically added to a final concentration of 10%. The plates were then returned to the incubator for 2 h. The medium was totally removed and cells and formazan crystals were solubilized by trituration in 400 µL of DMSO per well. The optical density of each well was spectrophotometrically measured at 570 nm. All experiments were repeated with at least four separate batches of cultures.

### 4.7. Cell Phenotypes

Cell phenotypes were revealed by immunochemistry. In adherent conditions, medium was removed from the wells, and cells were washed with PBS before being fixed with 4% paraformaldehyde (PFA) in PBS for 10 min. Cells were washed again and treated with 0.1% Triton in PBS for 20 min. After another washing procedure, cells were treated with 1% bovine serum albumin (BSA) to block unspecific binding sites. Cells were incubated with primary antibodies for 2 h at room temperature (Table 1). Finally, after several washing procedures, the second antibodies (conjugated to Alexa-488 or -555, Molecular probes) were incubated for 2 h and cells were counterstained with diamidino phenylindole (DAPI, 5 µg/mL) for 2 min.

In nonadherent conditions (i.e., neurosphere culture), the same procedure was applied for immunochemistry at different culture times (2, 7, 14, 21 days), except that each reagent addition or washing step was followed by centrifugation at 100× *g* for 8 min. In addition, cells were dissociated with accutase (Sigma; 8 min at 37 °C) before the whole procedure. It was especially important to assure a correct counting of cell phenotypes in neurospheres where cells appear in compact aggregates. We nevertheless also labeled neurospheres without dissociation, but only to show images corresponding to reality and not for counting (see Figure 3). All phenotype counts were obtained after a homogenization at the last step of the labeling procedure using a 100 µL droplet laid on a slide. Phenotype percentages were calculated after cell counting under a microscope calibrated in micrometers.

## Figures and Tables

**Figure 1 ijms-21-07249-f001:**
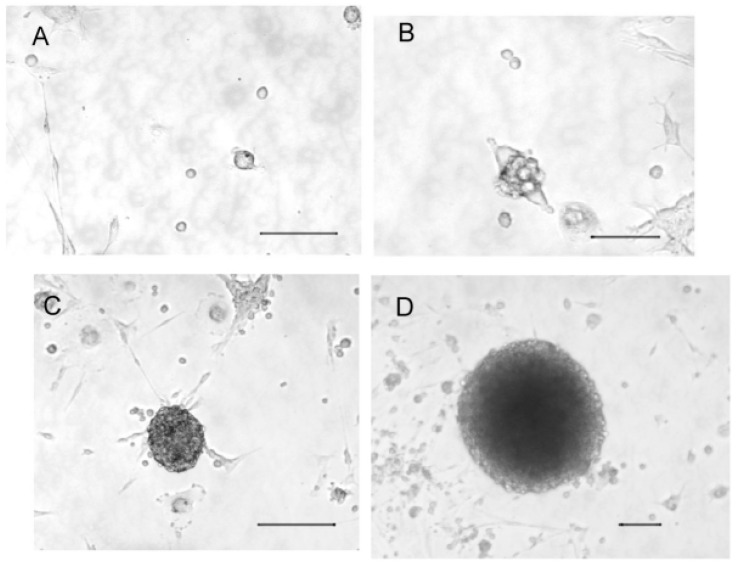
Phase-contrast microscopy of olfactory progenitor cultures after 1, 5, 10, and 21 days ((**A**–**D**) respectively; bars represent 50 µm).

**Figure 2 ijms-21-07249-f002:**
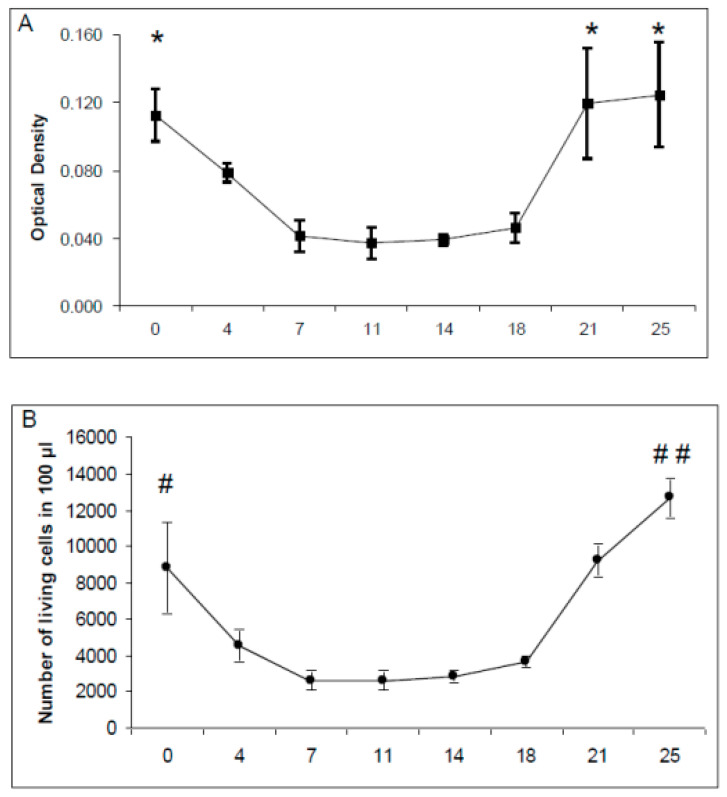
Cell viability. (**A**) dimethylthiazol-diphenyl tetrazolium salt (MTT) colorimetric measurement. (**B**) Living cells counting with trypan blue exclusion. ANOVA summary: * significantly higher than measures after 4 to 18 days with F (7-32) = 23.7; *p* < 0.0001. # significantly higher than measures after 4 to 18 days with F (7-24) = 48.1; *p* < 0.0001. ## significantly higher than all other measures with F(7-24) = 48.1; *p* < 0.0001.

**Figure 3 ijms-21-07249-f003:**
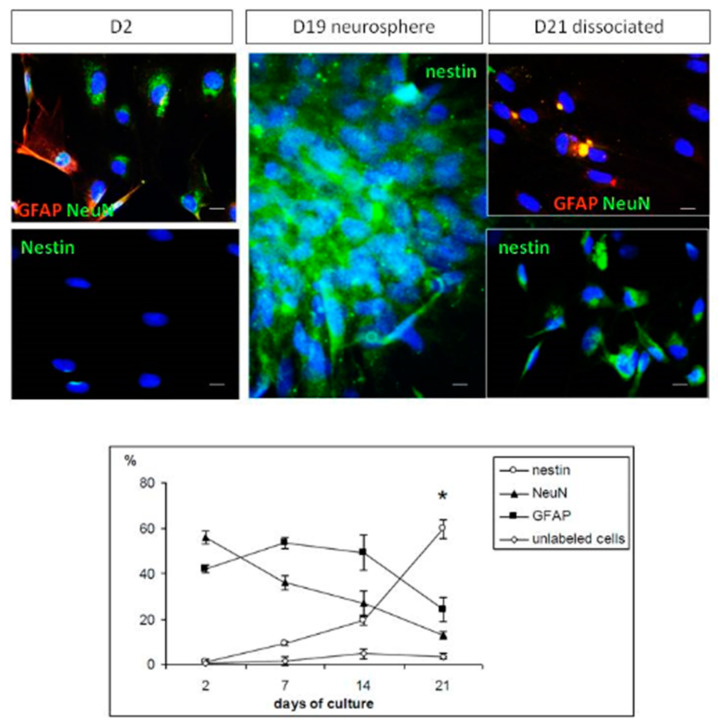
Time course of cell phenotypes during the first three weeks of culture of olfactory epithelia. Upper panel: immunocytochemistry for Glial Fibrillary Acidic Protein (GFAP), Neuronal Nuclear protein (NeuN), and Nestin markers on days 2, 19 and 21. Images on days 19 and 21 are related since cell counting was difficult in compact neurospheres (day 19); thus, neurospheres were dissociated for counting (day 21). Note that the dissociation procedure could alter the quality of images obtained after classical immunocytochemistry (images on day 21 compared to others); 20× magnification; bars represent 10 µm. Lower panel: cell counting; *t*-test analysis: * proportion significantly higher on day 21 compared to day 2 for Nestin-positive cells (*t* = −27.2, *p* < 0.0001) and proportions significantly lower on day 21 compared to day 2 for NeuN- and GFAP-positive cells (*t* = 6.27 to 25.9, *p* < 0.0008).

**Figure 4 ijms-21-07249-f004:**
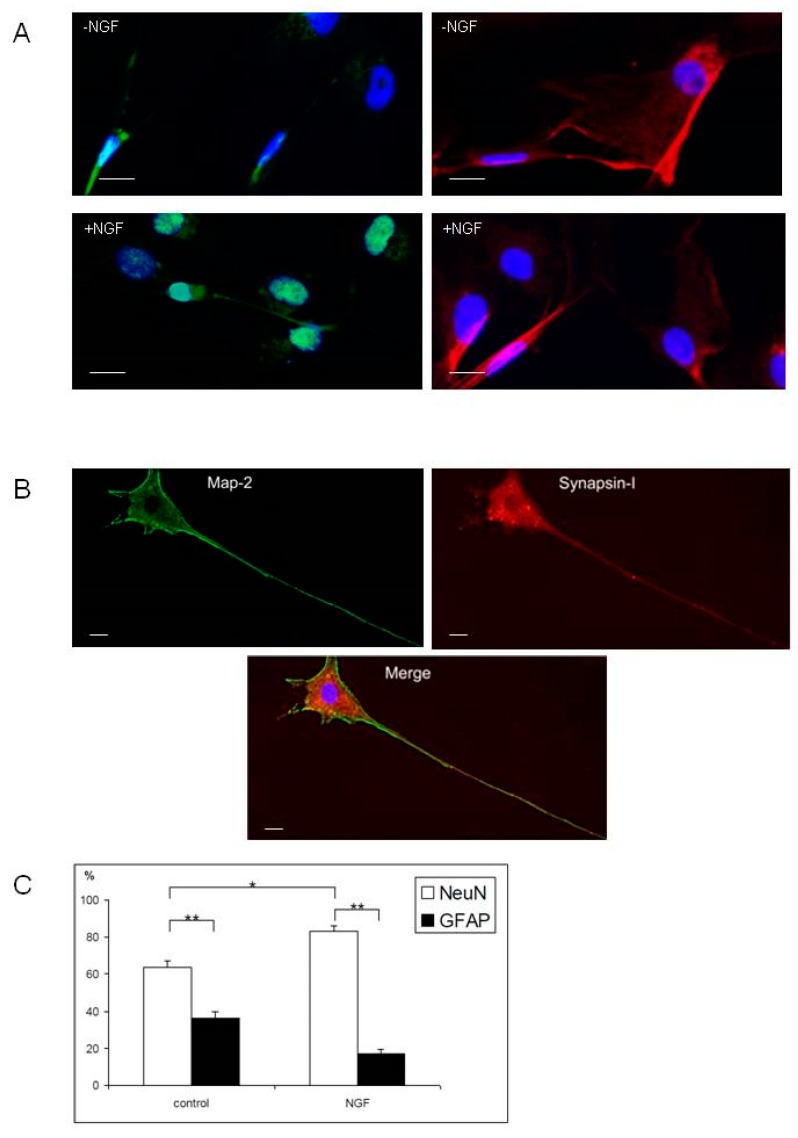
Cell phenotypes after 10 days of culture of olfactory progenitors in adherent condition. (**A**) Neuronal differentiation. (**B**) Glial differentiation. (**C**) Neuronal differentiation with the expression of synaptogenic marker; 40× and 20× magnification for (**A**,**B**) and (**C**), respectively; bars represent 10 µm. (**D**) Proportion of cells expressing neuronal (NeuN) and glial (GFAP) markers in media with or without Nerve Growth Factor (NGF); *t*-test analysis: * proportion significantly different (*t* = 9.9, *p* < 0.0001). ** proportions significantly different (*t* = 12.4 to 40.0, *p* < 0.0001).

**Figure 5 ijms-21-07249-f005:**
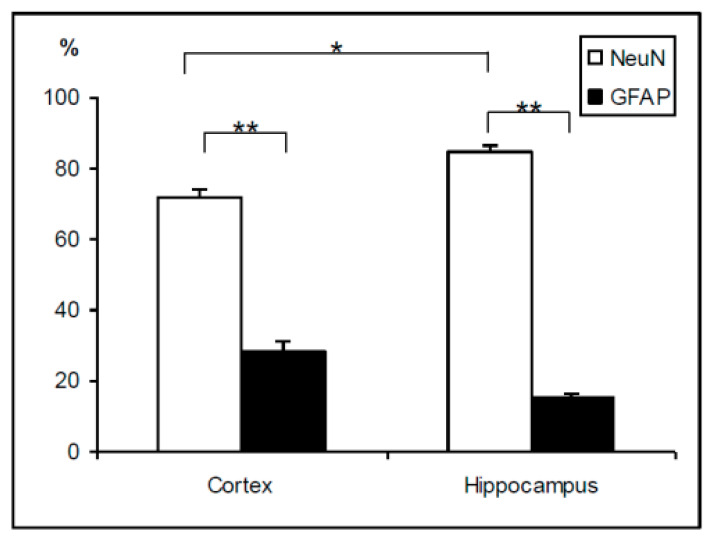
Proportions of cells expressing neuronal (NeuN) and glial (GFAP) markers in primary culture of the cortex and hippocampus after 6 days; *t*-test analysis: * proportion significantly different (*t* = 3.7, *p* < 0.003). ** proportions significantly different (*t* = 11.0 to 24.6, *p* < 0.0001).

**Figure 6 ijms-21-07249-f006:**
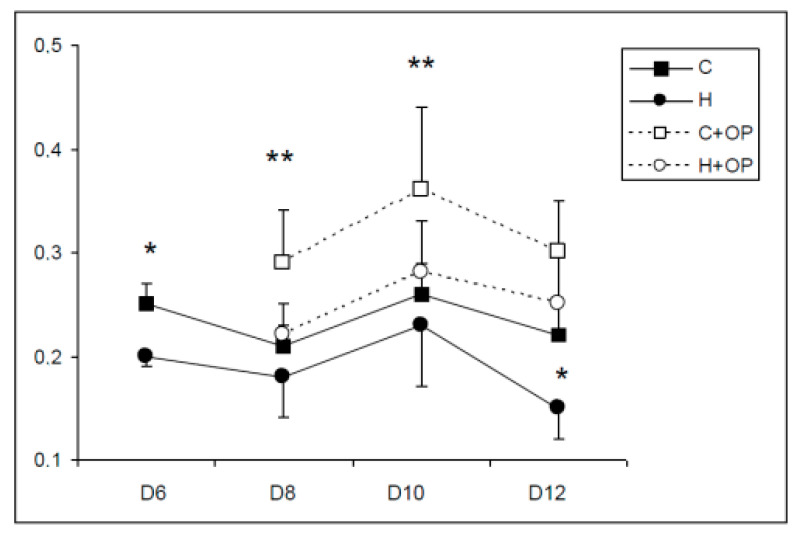
Cell viability measured using the MTT method in primary and cocultures. Values are optical densities (means ± standard error (SE)). ANOVA analysis: * cortex values are significantly higher than hippocampus (F(1-14) = 9.5 to 9.7, *p* < 0.008). ** cortex + olfactory progenitor values are significantly higher than cortex (F(3-28) = 2.8 to 3.9, *p* < 0.05).

**Figure 7 ijms-21-07249-f007:**
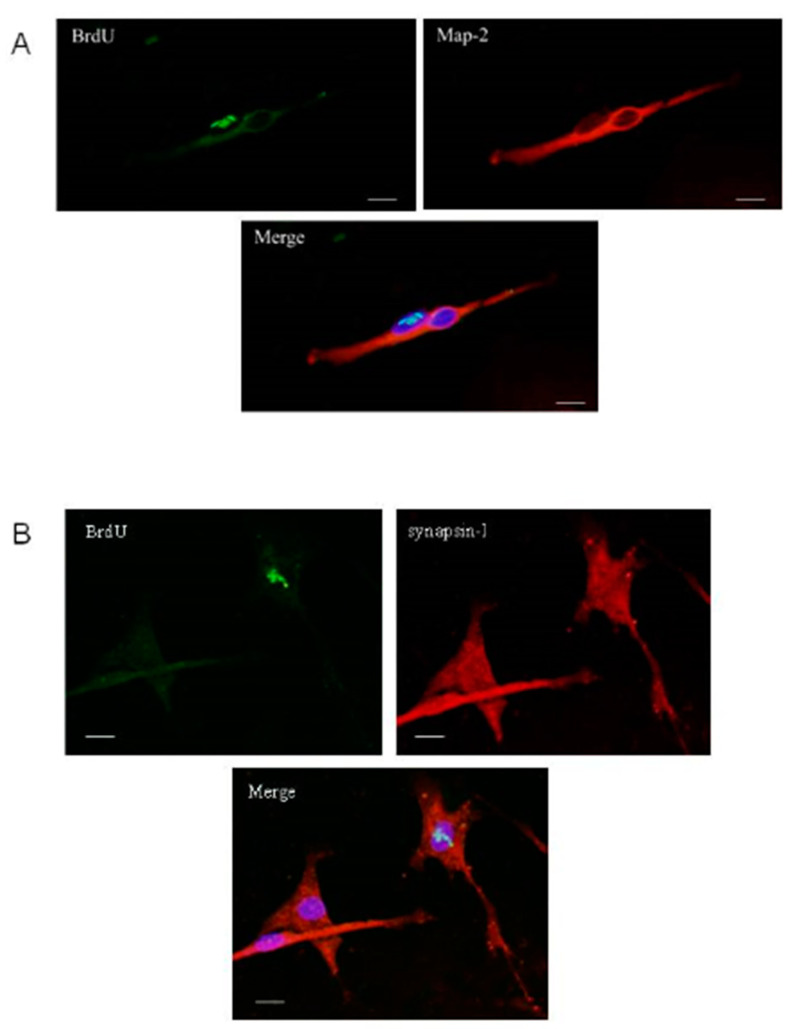
Cell phenotypes on day 12 of coculture. Pictures show cells originated from olfactory progenitors (BrdU-labeled) and cells originated form primary cultures (not BrdU-labeled). (**A**) Olfactory progenitor coexpresses BrdU and Map-2 markers. (**B**) Olfactory progenitor coexpresses BrdU and synapsin-I markers; 40× magnification; bars represent 10 µm.

**Table 1 ijms-21-07249-t001:** List of used antibodies, suppliers, and working utilization.

Primary Antibodies	Supplier	Working Dilution	Phenotype
Nestin	Chemicon	1/300	Undifferentiated cells
GFAP	Sigma	1/200	Astrocytes
NeuN	Chemicon	1/300	Mature neurons
Map-2	Chemicon	1/400	Cytoplasmic and axonal microtubules in neurons
Synapsin-I	Chemicon	1/400	Cytoplasmic and axonal synaptic proteins in neurons

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
