# Peer review of "The Fate of Transplanted Olfactory Progenitors Is Conditioned by the Cell Phenotypes of the Receiver Brain Tissue in Cocultures"

_ijms, 2020, doi:10.3390/ijms21197249_

Round 1

Reviewer 1 Report

This paper suggest that the integration of transplanted olfactory progenitor might by affected by both trophic factors and the environmental conditions/cell phenotypes of the host tissue. Thus, a model of co-culture could provide useful informations on key cell events for the use of progenitors in cell therapy. And I athink this paper is interesting and valuable to accept. 

And English language and style are fine/minor spell check required.

line 19 were -> was

line 20 an ->a

line 187 have -> had

line 187 of -> for

line 206 adult -> adults

line 211 min -> mins

Author Response

Thank you very much for you help.

Your proposed modifications are in the revised manuscript.

Sincerely

Reviewer 2 Report

Comments to authors:

This paper studies the behavior of olfactory progenitors in culture and their behavior while co-cultured in vitro with cortical and hippocampal neurons. The new data reported here for OPs behavior in vitro is very very limited. I will consider the publication of this paper after addressing the following comments:

Major comments:

  • Figure 3 needs to be completed with a panel of images showing the immunocytochemistry results for the different markers (like Fig. 4), at least at days 2 and 21.
  • Figure 3: authors must include a section in Materials and Methods describing on detail how cells quantifications were made.
  • Page 5, lines 101-102: what do authors mean by “to differentiate by themselves”?
  • Figure 4: images should show both neuronal and glial markers in both culture conditions. Also, images from panels A and B should be of the same size. Take the time to edit this figure appropriately. An image showing the DAPI alone is unnecessary.
  • Figure 6: results shown in this figure need to be clearly explained within the text. Thus, in the first paragraph on page 8 (lines 141-144), authors report an increase on cell viability in all scenarios, although it is better on co-cultures over cortical neurons. Which cells best increased their viability: the neurons in culture or the OPs seeded on the top? Because the interpretation of the results could be completely different.
  • Section 2.5 and Fig. 7: authors speculate that the differentiation of OPs in co-culture giving rise only to neurons, was caused because “…progenitors should respond to the differentiating factor NGF by producing only neuronal cells” (Page 8, line 151). If I am not wrong and after reading Materials and Methods, after the OPs were plated on top of the neuronal cultures, NGF was not added to the culture so, how can NGF be playing a role in OPs differentiation? And, if NGF was added to the co-cultures, what was its effect on the neurons already in the plate?

Minor comments:

  • Page 1, line 20: “…indicating an healthy…” delete “n” in “an”
  • Page 2, line 27: replace “Since” by “During”
  • Page 2, lines 34-35: “…to avoid donor destruction…” What do you mean by that?
  • Page 2, line 37, replace sentence by: However, important ethic problems need to be addressed before considering any of these approaches for clinical applications.”
  • Page 2, line 39, replace by: “Accessible sources of progenitor cells are…”
  • Page 2, line 42, replace by: “Thus, olfactory progenitors (OP) can be easily…” / Line 43: delete “can be” after [29]; “…and they are suitable for…”
  • Page 2, lines 44-45, replace: “not only for cell replacement strategies” by “not only for strategies of cell replacement”
  • Page 2, line 51: replace “…and they…” by “…and their...”
  • Page 2, line 55: replace “graft” by “grafting”
  • Page 2, line 57: “Therefore, we developed…”
  • Page 2, line 61: replace “enlargement” by “magnification”
  • Page 2, line 63: “…of culture, cells formed larger aggregates, and after 2 weeks they grew as neurospheres…”
  • Page 3, line 69: “…in cultures of OPs”
  • Page 3, line 69: “Both methods...” describe the methods here
  • Page 3, lines 75-76: write: “The number of living cells was incremented by 33% while compared with the beginning of the study, showing a significant increase from DAY X to DAY X….
  • Page 4, lines 84-85: we followed their maturation during the first 3 weeks by immunocytochemistry…”
  • Page 4, line 88: delete “The”
  • Page 4, line 89: rewrite as “…where immature cell phenotypes were undetected.”
  • Page 4, line 93: rewrite as “an increase in the number of immature cells …..by day 21”
  • Page 5, line 97: replace olfactory epithelium by olfactory progenitors
  • Page 5, line 106: move “respectively” at the end of the sentence.
  • Page 5, line 108: replace “…markers of neuron” by “markers of neuronal”
  • Page 6, line 120: replace “increased” by “promoted”
  • Page 6, line 124: delete “cell phenotypes corresponding to”
  • Page 6, line 127: delete “ones”
  • Page 8, line 141: replace “on such” by “on top”
  • Page 9, line 158: Olfactory
  • Page 10, line 221: delete one “of culture”
  • Page 11, line 268: replace “to coat” by “to block”

Author Response

This paper studies the behavior of olfactory progenitors in culture and their behavior while co-cultured in vitro with cortical and hippocampal neurons. The new data reported here for OPs behavior in vitro is very very limited. I will consider the publication of this paper after addressing the following comments:

General response to reviewer2:

Thank you very much for you review.

We agree with you concerning the "new data" on OPs in the present paper. The first part is not really new. We did not focus our article on OPs culture/production. As mentioned in the introduction, it is known for OPs and other sources of progenitors since several years now. Nevertheless, we had to explain and prove how we obtained, produced and differentiated OPs, as the beginning of the "story". Several authors of this paper are implicated in cell culture since more than 20 years; and in the 90's and the beginning of the century, it was a general run to produce neurons and to try to replace neurons. Very little studies were focused on glial cells or on neuron-glia crosstalk. Nowadays it is a little bit different, and new studies emerge with a topic concerning the importance of glial cell for the fate of newly produced/transplanted neurons. We wrote our discussion in that sense; especially the second and last part. To our mind the most important message of the present paper is to enhance further studies to focus on "friendly environments", including glial cells, for the fate of graft cells/neurons. See references 53, 54 and 57 - 59 published 2017 - 2019. If necessary, we can modify the discussion to reinforce the message.

Nevertheless, we modified the manuscript as you proposed in the major and minor comments.

Major comments:

  • Figure 3 needs to be completed with a panel of images showing the immunocytochemistry results for the different markers (like Fig. 4), at least at days 2 and 21. The figure is completed with day 2, day 19 (only to show how nestin reactive are the neurospheres), and day 21 after dissociation of neurospheres for counting. For this last case (day 21 labeling), the quality of immunocytochemistry appears a little bit altered due to the cells dissociation and immediate labeling. A precision for that is added in the figure legend, including the magnification and scale-bars.
  • Figure 3: authors must include a section in Materials and Methods describing on detail how cells quantifications were made. Precisions are made at the end of the 4.7 sub-section "non-adherent conditions ... and counting".
  • Page 5, lines 101-102: what do authors mean by “to differentiate by themselves”? We agree; this is not clear and confusing. Modified by " In order to investigate whether cells dissociated from neurospheres were able to differentiate properly in the neuronal lineage, ..."
  • Figure 4: images should show both neuronal and glial markers in both culture conditions. As mentioned in the 2.3. sub-section of the results, " Qualitatively, both media allowed cells to differentiate in neurons or astrocytes, shown by immunoreactivity to NeuN and GFAP respectively (Figure 4)". Thus, if we add all conditions in the figure, images will look like the same for both conditions (with or without NGF). If it is still needed, we can add more images (editor choice?). Also, images from panels A and B should be of the same size. Take the time to edit this figure appropriately. An image showing the DAPI alone is unnecessary. We modified the figure, deleting DAPI panels and adding bars for the scale (in the figure legend too). We kept the C-panels unmodified because to our mind the long labeled axon reinforce our argumentation and the calibrated bars allow the visual comparison (editor choice?).
  • Figure 6: results shown in this figure need to be clearly explained within the text. Thus, in the first paragraph on page 8 (lines 141-144), authors report an increase on cell viability in all scenarios, although it is better on co-cultures over cortical neurons. Which cells best increased their viability: the neurons in culture or the OPs seeded on the top? Because the interpretation of the results could be completely different. Using the MTT method, it is only possible to monitor the global viability of a dish. Nevertheless considering that cell densities are controlled since the first day of culture, it is possible to discuss and propose a conclusion. We modified the manuscript in that sense (end of 2.4. sub-section).
  • Section 2.5 and Fig. 7: authors speculate that the differentiation of OPs in co-culture giving rise only to neurons, was caused because “…progenitors should respond to the differentiating factor NGF by producing only neuronal cells” (Page 8, line 151). If I am not wrong and after reading Materials and Methods, after the OPs were plated on top of the neuronal cultures, NGF was not added to the culture so, how can NGF be playing a role in OPs differentiation? And, if NGF was added to the co-cultures, what was its effect on the neurons already in the plate? All our apologizes for that. It was not written properly. Of course, NGF was not include in co-cultures. The sentence is modified in the revised version (end of 2.5. sub-section).

Minor comments: All corrected in the revised version

  • Page 1, line 20: “…indicating an healthy…” delete “n” in “an”
  • Page 2, line 27: replace “Since” by “During”
  • Page 2, lines 34-35: “…to avoid donor destruction…” What do you mean by that?
  • Page 2, line 37, replace sentence by: However, important ethic problems need to be addressed before considering any of these approaches for clinical applications.”
  • Page 2, line 39, replace by: “Accessible sources of progenitor cells are…”
  • Page 2, line 42, replace by: “Thus, olfactory progenitors (OP) can be easily…” / Line 43: delete “can be” after [29]; “…and they are suitable for…”
  • Page 2, lines 44-45, replace: “not only for cell replacement strategies” by “not only for strategies of cell replacement”
  • Page 2, line 51: replace “…and they…” by “…and their...”
  • Page 2, line 55: replace “graft” by “grafting”
  • Page 2, line 57: “Therefore, we developed…”
  • Page 2, line 61: replace “enlargement” by “magnification”
  • Page 2, line 63: “…of culture, cells formed larger aggregates, and after 2 weeks they grew as neurospheres…”
  • Page 3, line 69: “…in cultures of OPs”
  • Page 3, line 69: “Both methods...” describe the methods here. The methods are described in the "Methods section" so is it necessary to technically describe here the methods? We propose a new sentence where the methods used are cited " Both used methods (i.e. MTT and Trypan Blue exclusion) showed the same profile during the culture (Figure 2)"
  • Page 3, lines 75-76: write: “The number of living cells was incremented by 33% while compared with the beginning of the study, showing a significant increase from DAY X to DAY X….
  • Page 4, lines 84-85: we followed their maturation during the first 3 weeks by immunocytochemistry…”
  • Page 4, line 88: delete “The”
  • Page 4, line 89: rewrite as “…where immature cell phenotypes were undetected.”
  • Page 4, line 93: rewrite as “an increase in the number of immature cells …..by day 21”
  • Page 5, line 97: replace olfactory epithelium by olfactory progenitors
  • Page 5, line 106: move “respectively” at the end of the sentence.
  • Page 5, line 108: replace “…markers of neuron” by “markers of neuronal”
  • Page 6, line 120: replace “increased” by “promoted”
  • Page 6, line 124: delete “cell phenotypes corresponding to”
  • Page 6, line 127: delete “ones”
  • Page 8, line 141: replace “on such” by “on top”
  • Page 9, line 158: Olfactory
  • Page 10, line 221: delete one “of culture”
  • Page 11, line 268: replace “to coat” by “to block”

Round 2

Reviewer 2 Report

Some of my previous comments are not addressed in the revised version, so I will consider the acceptance of this paper only when one and each of my comments and questions are properly addressed. Also, please highlight all modification made in the text with a different color.

My original comments are in black, answers from authors are in red, and my new comments are in blue/bold:

  • Figure 3 needs to be completed with a panel of images showing the immunocytochemistry results for the different markers (like Fig. 4), at least at days 2 and 21. The figure is completed with day 2, day 19 (only to show how nestin reactive are the neurospheres), and day 21 after dissociation of neurospheres for counting. For this last case (day 21 labeling), the quality of immunocytochemistry appears a little bit altered due to the cells dissociation and immediate labeling. A precision for that is added in the figure legend, including the magnification and scale-bars. Immunohistochemistry panels are still missing from the revised version of the manuscript
  • Figure 3: authors must include a section in Materials and Methods describing on detail how cells quantifications were made. Precisions are made at the end of the 4.7 sub-section "non-adherent conditions ... and counting". The corrected text is still incomplete. In my initial revision I was requesting to describe the quantification method, explaining on detail information such as the magnification used to take the images, the number of fields per slide that were quantified, how many animals (N) were used to get the cells for quantifications, if replicas of the same experiment were made, and the stereological method used for these quantifications
  • Page 5, lines 101-102: what do authors mean by “to differentiate by themselves”? We agree; this is not clear and confusing. Modified by " In order to investigate whether cells dissociated from neurospheres were able to differentiate properly in the neuronal lineage, ..."
  • Figure 4: images should show both neuronal and glial markers in both culture conditions. As mentioned in the 2.3. sub-section of the results, " Qualitatively, both media allowed cells to differentiate in neurons or astrocytes, shown by immunoreactivity to NeuN and GFAP respectively (Figure 4)". Thus, if we add all conditions in the figure, images will look like the same for both conditions (with or without NGF). If it is still needed, we can add more images (editor choice?). Even if cells look identical in both conditions, a representative image of the results on both media should be shown, as they both were studied and results described in this paper. Including or not additional figures, as well as requesting additional experiments, is a reviewer but not an editor decision.  
  • Also, images from panels A and B should be of the same size. Take the time to edit this figure appropriately. An image showing the DAPI alone is unnecessary. We modified the figure, deleting DAPI panels and adding bars for the scale (in the figure legend too). We kept the C-panels unmodified because to our mind the long labeled axon reinforce our argumentation and the calibrated bars allow the visual comparison (editor choice?). The new figure 4 is identical to the one in the original submission.
  • Figure 6: results shown in this figure need to be clearly explained within the text. Thus, in the first paragraph on page 8 (lines 141-144), authors report an increase on cell viability in all scenarios, although it is better on co-cultures over cortical neurons. Which cells best increased their viability: the neurons in culture or the OPs seeded on the top? Because the interpretation of the results could be completely different. Using the MTT method, it is only possible to monitor the global viability of a dish. Nevertheless considering that cell densities are controlled since the first day of culture, it is possible to discuss and propose a conclusion. We modified the manuscript in that sense (end of 2.4. sub-section). I am very familiar with the MTT method and using MTT was not a concern to me. I asked about the interpretation of the results, and as far as I can tell, in this revised version my question is still unanswered. Therefore, how can authors know that the increase in viability in cocultures of OPs and cortical neurons is caused by a reduced mortality of the OPs (page 9, line153-154)? If you want to analyze survival/mortality of OPs (or any cell), then additional and alternative methods need to be used to support that statement.

Author Response

Please find here the details of the modifications given in the revised version.

  • Figure 3 needs to be completed with a panel of images showing the immunocytochemistry results for the different markers (like Fig. 4), at least at days 2 and 21. The figure is completed with day 2, day 19 (only to show how nestin reactive are the neurospheres), and day 21 after dissociation of neurospheres for counting. For this last case (day 21 labeling), the quality of immunocytochemistry appears a little bit altered due to the cells dissociation and immediate labeling. A precision for that is added in the figure legend, including the magnification and scale-bars. Immunohistochemistry panels are still missing from the revised version of the manuscript. I suppose that the reviewer did not receive the correct modified figures because I uploaded a ZIP format, since the on-line interface does not allow to transfer more than one file for figures (JPEG format for example). Hope to succeed for this second revision.
  • Figure 3: authors must include a section in Materials and Methods describing on detail how cells quantifications were made. Precisions are made at the end of the 4.7 sub-section "non-adherent conditions ... and counting". The corrected text is still incomplete. In my initial revision I was requesting to describe the quantification method, explaining on detail information such as the magnification used to take the images, the number of fields per slide that were quantified, how many animals (N) were used to get the cells for quantifications, if replicas of the same experiment were made, and the stereological method used for these quantifications. More precisions are given in the 4.7. sub-section, highlighted in yellow.
  • Figure 4: images should show both neuronal and glial markers in both culture conditions. As mentioned in the 2.3. sub-section of the results, " Qualitatively, both media allowed cells to differentiate in neurons or astrocytes, shown by immunoreactivity to NeuN and GFAP respectively (Figure 4)". Thus, if we add all conditions in the figure, images will look like the same for both conditions (with or without NGF). If it is still needed, we can add more images (editor choice?). Even if cells look identical in both conditions, a representative image of the results on both media should be shown, as they both were studied and results described in this paper. Including or not additional figures, as well as requesting additional experiments, is a reviewer but not an editor decision. It was just a simple question for the editor in order to eventually save space or reduce figure size, and not a criticism of the reviewer position. I am also a reviewer and an editor and I am usual with such subjects. Additional images are given in figure 4 for both conditions (+/- NGF). Don't know if the negative labeling (black images) for both GFAP and NeuN markers are needed in panel A.
  • Also, images from panels A and B should be of the same size. Take the time to edit this figure appropriately. An image showing the DAPI alone is unnecessary. We modified the figure, deleting DAPI panels and adding bars for the scale (in the figure legend too). We kept the C-panels unmodified because to our mind the long labeled axon reinforce our argumentation and the calibrated bars allow the visual comparison (editor choice?). The new figure 4 is identical to the one in the original submission. One more time I suppose that the on-line transfer failed for the revised figure 4. We proposed a revised figure 4 with calibration/scale bars in order to conserve the interesting long view of the labeled axon typically representing a neuronal shape in new panel B (panel C of the initial version). All images are now of the same size.
  • Figure 6: results shown in this figure need to be clearly explained within the text. Thus, in the first paragraph on page 8 (lines 141-144), authors report an increase on cell viability in all scenarios, although it is better on co-cultures over cortical neurons. Which cells best increased their viability: the neurons in culture or the OPs seeded on the top? Because the interpretation of the results could be completely different. Using the MTT method, it is only possible to monitor the global viability of a dish. Nevertheless considering that cell densities are controlled since the first day of culture, it is possible to discuss and propose a conclusion. We modified the manuscript in that sense (end of 2.4. sub-section). I am very familiar with the MTT method and using MTT was not a concern to me. I asked about the interpretation of the results, and as far as I can tell, in this revised version my question is still unanswered. Therefore, how can authors know that the increase in viability in cocultures of OPs and cortical neurons is caused by a reduced mortality of the OPs (page 9, line153-154)? If you want to analyze survival/mortality of OPs (or any cell), then additional and alternative methods need to be used to support that statement. Using "mortality" could be consider as speculation (removed). Nevertheless, we carefully controlled the cell amounts at both critical points of cultures; the early beginning of the primary cultures and the day of co-culture (day 8). This is modified (in yellow) at the end of the 2.4. sub-section. Thus we are able to discuss this with respect to the literature. Such a discussion is modified in the related paragraph of the discussion section (in yellow), just before linking our results to the literature [51-54]. We carefully chose new words to avoid speculation but to support our finding with convergent results in other comparable studies also using co-cultures.

Round 3

Reviewer 2 Report

All my comments and concerns have been addressed properly and this paper is now ready for publication.